# INSTEMB: INSTRUCTION-FOLLOWING EMBEDDINGS THROUGH LOOK-AHEAD TOKEN DISTILLATION

## ABSTRACT

Recent advances have empowered large language models (LLMs) with remarkable fine-grained instruction-following capabilities in text generation tasks. However, embedding methods typically rely solely on the hidden state of the input's last token, limiting their ability to capture complete semantic signals distributed across the full output tokens. Moreover, existing discrete-to-continuous re-encoding approaches introduce semantic discontinuity. To address these limitations, we propose **InstEmb**, a novel instruction following embedding framework. InstEmb jointly optimizes two key aspects: (1) primary semantic information, achieved by employing contrastive learning focused on the representation of the last input token, and (2) complementary semantic information, captured through representation distillation leveraging learnable look-ahead tokens without introducing additional decoding latency. Additionally, we introduce **Dual-Anchor Alignment Pooling (DAAP)**, explicitly aligned with our dual training objectives. Extensive experiments demonstrate that InstEmb achieves state-of-the-art performance across multiple instruction following benchmarks without benchmark-specific supervised data.

## 1 INTRODUCTION

Recent advances in large language models (LLMs) have significantly enhanced their capability to follow fine-grained instructions, enabling remarkable zero-shot performance across diverse downstream tasks (Wang et al., 2023b; Shen et al., 2023; Ji et al., 2023; Zhang et al., 2023; Naveed et al., 2023). While instruction-following abilities have been extensively leveraged in text generation tasks, it remains challenging to achieve similar fine-grained instruction adaptability when utilizing LLMs to produce text embeddings. Ideally, embeddings generated under different instructions should capture distinct semantic aspects of the same input text, providing richer and more targeted semantic representations (Li et al., 2024a; Weller et al., 2024a; Li et al., 2024b; Weller et al., 2024b).

For instance, in product retrieval, embedding methods face challenges with queries exhibiting minor semantic variations. Consider two similar queries for a tent: "*Is this tent durable for outdoor use?*" and "*Is this tent compact for outdoor use?*". While both concern outdoor suitability, their emphasis differs—durability versus portability. Embeddings should reflect these nuances within the product detail. Critically, while traditional encoders struggle when queries share high semantic overlap, due to their inability to capture fine-grained instruction shifts. This limits their effectiveness in instruction following retrieval.

Prior research indicated that aligning the embedding space with key tokens enhances semantic interpretability and instruction-following performance. Specifically, (Nie et al., 2024; Yamada & Zhang, 2025) demonstrated that LLM-derived embeddings inherently exhibit strong alignment with key tokens, while (Peng et al., 2024) showed that fine-tuning towards key tokens further improves semantic quality.Although previous works have shown the last token pooling approach achieves strong performance (Tang & Yang, 2024; Peng et al., 2024), it fundamentally conflates distinct semantic roles. The last token primarily captures the semantics of the input and instruction but fails to incorporate the semantic information of output, as the latter is distributed across multiple critical tokens and cannot be adequately aggregated into a single position. To clarify this distinction, we hereafter refer to these two roles as **primary semantics** and **complementary semantics.**

In the realm of Retrieval-Augmented Generation (RAG), recent attempts to enhance complementary semantics capabilities through re-encoding of output tokens or hypothetical documents (Gao et al.,

2023; Ma et al., 2023) introduce another limitation: The method enacts discrete-to-continuous conversion through autoregressive token generation followed by re-encoding. This dual-stage process creates a semantic reconstruction gap, where sampling during decoding severs continuity in latent representations.

These observations suggest that optimizing embedding results requires consideration of the following key aspects: 1) Both primary semantics and complementary semantics need to be used simultaneously, and the acquisition of complementary semantics should not incur additional time consumption due to decoding. 2) Both types of semantics should ideally be aggregated and preserved within the LLM's latent space, rather than relying on re-encoding or discrete-to-continuous mapping.

To address these limitations, we propose a novel instruction following embedding framework, **InstEmb**, Our method explicitly optimizes both types of information: primary semantics captured by the last input token, and complementary semantics distributed across the suffix of learnable special tokens, which we hereafter refer to as **look-ahead** tokens. We adopt this terminology because the function of our special tokens is similar to the concept of speculative decoding field (Monea et al., 2023; Xiao et al., 2024; An et al., 2025). The look-ahead tokens, which can be conceptually viewed as a form of suffix soft prompt, provide the model with implicit semantic previews of future tokens.

During inference, **InstEmb** generates embeddings solely through the prefilling stage, efficiently encoding output-related information without multi-step decoding. Furthermore, we propose **DAAP**, a pooling strategy explicitly aligned with our training objectives, eliminating empirical pooling selection and ensuring consistently optimal embedding performance.

We empirically validate InstEmb across multiple instruction following dense retrieval benchmarks, demonstrating strong effectiveness and efficiency compared to state-of-the-art baselines such as Inbedder (Peng et al., 2024) and FollowIR (Weller et al., 2024a).

In summary, our contributions are:

- We propose **InstEmb**, a novel instruction following embedding framework utilizing learnable look-ahead tokens and embedding distillation to achieve fine-grained, sentence-level instruction adaptation.

- We introduce a novel embedding distillation objective based on look-ahead tokens that effectively transfers instruction-following capabilities from a frozen instruction-tuned LLM.

- InstEmb attains state-of-the-art performance across multiple instruction following benchmarks without benchmark-specific supervised training.

## 2 RELATED WORK

**Instruction-tuned Embedding based on LLMs.** Recent embedding approaches leverage large language models (LLMs) by fine-tuning them with instructions to enhance semantic representation quality and adaptability. Recent works predominantly utilize fixed, task-level instructions: LLM2Vec (BehnamGhader et al., 2024) transforms autoregressive decoders into bidirectional encoders for stronger representational power; GRITLM (Muennighoff et al., 2024) unifies generative and representational tasks through instruction-tuning; E5-Mistral (Wang et al., 2023a) employs contrastive tuning on synthetic query-document pairs; NV-Embed (Lee et al., 2024) optimizes embeddings via a two-stage contrastive learning paradigm; and ECHO (Springer et al., 2024) repeats input contexts to mitigate autoregressive limitations. Instructor (Su et al., 2022), while adopting contrastive training across numerous NLP tasks, still uses fixed task-level instructions rather than dynamically adapting instructions per instance. Moving beyond task-level instructions, recent approaches dynamically adjust embeddings according to instance-level (per-query) instructions. BGE-icl (Li et al., 2024a) and RARE (Tejaswi et al., 2024) leverage in-context learning to adapt embeddings using task-specific few-shot examples; PIE (Li et al., 2024b) constructs contrastive examples guided by additional linguistic parsing; FollowIR (Weller et al., 2024a) employs professional assessor narratives to interpret complex search intents; Promptriever (Weller et al., 2024b) utilizes a large-scale, instance-level instruction dataset from MS MARCO; and INBEDDER (Peng et al., 2024) introduces an embed-via-answering paradigm, fine-tuning models on abstractive QA tasks conditioned on dynamic user instructions, significantly enhancing instruction-following capabilities. Another category of methods performs secondary mapping based on the raw embeddings output by LLMs under specific conditions. For

instance, GSTransformer(Feng et al., 2025) adapts pre-computed embeddings in real time to align with user instructions, guided by a small amount of text data with instruction-focused label annotation. Hyper-CL(Yoo et al., 2024) adapts sentence embeddings to various conditions by transforming pre-computed condition embeddings into corresponding projection layers. CASE(Zhang et al., 2025a) produces condition-aware sentence embeddings by first generating a context-informed condition embedding via attention interaction with the sentence, and then applying a supervised nonlinear projection.

**Prompt Tuning and Soft Prompts.**   Prompt tuning has emerged as an effective alternative to full-model fine-tuning, enabling efficient adaptation of embedding models by optimizing only a small number of learnable prompt tokens. Methods such as Prefix-Tuning (Li & Liang, 2021), Prompt Tuning (Lester et al., 2021), and P-Tuning v2 (Liu et al., 2021) demonstrate that soft prompts can achieve comparable or strong performance to full fine-tuning with significantly reduced computational overhead. Recent works extend this paradigm specifically to embedding tasks: PromptBERT (Jiang et al., 2022) choose prompt manually to enhance embedding quality for retrieval, while SPoT (Vu et al., 2021) transfers learned soft prompts from source tasks to initialize prompts for target tasks, enabling efficient prompt-based transfer learning. Similarly, SimPTC (Fei et al., 2022) integrates soft prompt tuning with contrastive learning, showing improved semantic alignment. These approaches underscore the potential of soft prompt-based tuning to efficiently encode task-specific semantic knowledge into embeddings.

**Knowledge Distillation**   Knowledge distillation has been widely employed to transfer semantic knowledge from powerful teacher models to smaller student embedding models. Early methods, such as DistilBERT (Sanh et al., 2019), TinyBert (Jiao et al., 2019), and MiniLM (Wang et al., 2020), primarily leverage logits or attention-based alignment. In addition, some contrastive distillation methods focus on optimizing representations, such as CRD (Tian et al., 2019), CKD (Xu et al., 2022), and SEED (Fang et al., 2021). These methods leverage contrastive loss functions to capture semantic relationships among data, thereby enhancing the representation learning capability of the student model.

## 3   METHODOLOGY

As discussed in the Introduction, existing embedding methods exhibit two fundamental limitations: (1) relying solely on the input last token's representation; (2) discrete-to-continuous re-encoding introduces a semantic reconstruction gap.

To overcome these limitations, **InstEmb** employs learnable look-ahead tokens appended directly to the input text, enabling continuous representation distillation from a frozen teacher model in order to get complementary semantic. It also leverages standard contrastive learning at the input last token position to maintain the performance.

### 3.1   ARCHITECTURE

Given an autoregressive LLM with input sequence $x$ (input text concatenated with instruction) and target output $y$, we define learnable look-ahead tokens $s = [s_1, s_2, \ldots, s_L]$ to form the training inputs:

$$x_{\text{student}} = [x; s], \quad x_{\text{teacher}} = [x; y_{\text{trunc}}]$$

where $Y_{\text{trunc}}$ represents the first (L) tokens of $Y$. Both student and teacher models are initialized from identical pretrained parameters, with the teacher frozen during training to enable the look-ahead tokens $S$ to model output sequence semantics through distillation. This architecture enables single-step inference without multi-step decoding overhead or semantic reconstruction gaps.

The overall architecture of InstEmb is illustrated in Figure 1, and we elaborate on each component in the following parts.

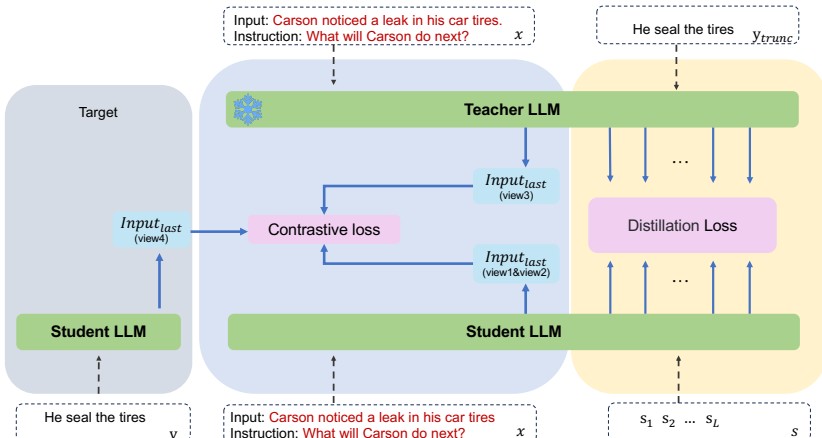

Figure 1: The overall architecture of InstEmb. It jointly optimizes two loss functions: a representation distillation loss to capture complementary semantic signals, and a contrastive learning loss at the input's last token position to enhance primary semantic alignment explicitly. The definition of every view could be found in 3.2.2.

## 3.2 TRAINING OBJECTIVES

To ensure a coherent flow of understanding, we begin by describing our approach to optimizing the complementary semantics.

### 3.2.1 REPRESENTATION DISTILLATION FOR COMPLEMENTARY SEMANTICS OPTIMIZATION

Inspired by knowledge distillation and speculative decoding (Li et al., 2024c; 2025), we employ position-wise distillation loss to align student look-ahead token representations with teacher target embeddings, enabling efficient transfer of complementary semantic information that is captured by the whole output sequence.

Formally, let $\boldsymbol{h}_i^S, \boldsymbol{h}_i^T \in \mathbb{R}^d$ denote the student and teacher hidden states at position $i$ ($i = 1, \ldots, |\boldsymbol{x}| + L$). We consider two distillation strategies: directly aligning hidden states (MSE) and aligning probability distributions (KL), to explore semantic alignment at different granularities.

- **Mean Squared Error (MSE) Loss:** Direct alignment of hidden states:

$$\mathcal{L}_{\text{Distill}} = \frac{1}{L} \sum_{i=|\boldsymbol{x}|+1}^{|\boldsymbol{x}|+L} \left\| \boldsymbol{h}_i^S - \boldsymbol{h}_i^T \right\|_2^2 \tag{1}$$

- **Kullback–Leibler (KL) Divergence Loss:** Alignment of output probability distributions obtained via language modeling heads:

$$\mathcal{L}_{\text{Distill}} = \frac{1}{L} \sum_{i=|\boldsymbol{x}|+1}^{|\boldsymbol{x}|+L} \sum_{v \in \mathbb{V}} P_i^S(v) \log \frac{P_i^S(v)}{P_i^T(v)} \tag{2}$$

where $P_i^S$ and $P_i^T$ denote the student and teacher distributions, respectively. $\mathbb{V}$ is the vocabulary set

From an information-theoretic perspective, the distillation loss maximizes the mutual information between student look-ahead representations and teacher target embeddings. This enables efficient knowledge transfer from the teacher's output-conditioned representations to the student's input-only accessible tokens. The position-wise alignment ensures that each look-ahead token captures specific aspects of the target sequence semantics, creating a distributed representation of complementary information that supplements the primary semantic signal at the input's last token.

### 3.2.2 Contrastive Learning for Primary Semantic Optimization

While representation distillation optimizes complementary semantics from look-ahead tokens, the dominant enhancement stems from primary semantic alignment at the last input token. Recent embedding optimization methods BehnamGhader et al. (2024); Springer et al. (2024) commonly utilize InfoNCE loss (Oord et al., 2018) for this purpose.

However, preliminary experiments revealed that traditional two-tower contrastive frameworks are suboptimal. We therefore introduce multiple embedding views with Supervised Contrastive Loss (Khosla et al., 2020), which generalizes InfoNCE to support multiple positive samples per anchor, enhancing training stability. For each instance, we construct four types of view:

- **Student Input (View 1&2)**: Two embeddings at position $|\boldsymbol{x}|$ from the student model, obtained via different dropout masks (SimCSE-style (Gao et al., 2021)).
- **Teacher Input (View 3)**: Embedding at position $|\boldsymbol{x}|$ from the teacher model, which is used to align with the distillation target.
- **Student Output (View 4)**: Embedding at position $|\boldsymbol{y}|$ from the student model with input $\boldsymbol{y}$, explicitly encoding output semantics.

For a minibatch of $N$ instances, the contrastive loss is:

$$\mathcal{L}_{\text{CL}} = -\frac{1}{N}\sum_{i=1}^{N}\frac{1}{|\mathbb{P}_i|}\sum_{z \in \mathbb{P}_i}\log\frac{\sum_{z^+ \in \mathbb{P}_i \setminus z}\exp(\frac{\text{sim}(z,z^+)}{\tau})}{\sum_{z' \in \mathbb{A} \setminus z}\exp(\frac{\text{sim}(z,z')}{\tau})}, \tag{3}$$

where $\mathbb{P}_i$ denotes the set of embedding views for instance $i$, $\mathbb{A} = \bigcup_{j=1}^{N}\mathbb{P}_j$ denotes all embeddings in the minibatch, $\text{sim}(\cdot,\cdot)$ is cosine similarity and $\tau$ is a temperature hyperparameter.

### 3.2.3 Joint Optimization

The final training objective jointly optimizes both complementary and primary semantic signals:

$$\mathcal{L}_{\text{InstEmb}} = \mathcal{L}_{\text{Distill}} + \mathcal{L}_{\text{CL}} \tag{4}$$

### 3.3 Dual-Anchor Alignment Pooling (DAAP)

InstEmb jointly optimizes primary semantics via contrastive learning at the last input token and complementary semantics via representation distillation on look-ahead tokens. However, standard pooling methods(e.g., last-token pooling or mean pooling) fail to explicitly integrate both optimized signals, capturing only a single semantic perspective.

To ensure theoretical alignment between training objectives and inference-time embedding extraction, we propose **Dual-Anchor Alignment Pooling (DAAP)**, which is explicitly designed to fuse two complementary semantic anchors without the need for empirical pooling selection:

- **Primary Semantic Anchor**: The hidden state of the final input token.
- **Complementary Semantic Anchor**: The averaged hidden states of the look-ahead tokens.

Formally, DAAP computes the final embedding as follows:

$$\boldsymbol{e} = \frac{1}{2}\left(\boldsymbol{h}_{|\boldsymbol{x}|}^{S} + \frac{1}{L}\sum_{j=|\boldsymbol{x}|+1}^{|\boldsymbol{x}|+L}\boldsymbol{h}_j^S\right) \tag{5}$$

## 4 Experiments

In this section, we empirically validate the effectiveness of our proposed InstEmb framework. We first introduce our experimental setup, including datasets, baselines, and evaluation metrics. Then, we present our main results, demonstrating the strength of InstEmb over existing state-of-the-art methods. Finally, we conduct extensive ablation studies to analyze the impact of different design choices.

## 4.1 Experimental Setup

### 4.1.1 Datasets

We utilize the training dataset proposed by Peng et al. (2024), comprising approximately 200,000 abstractive question-answer pairs from 11 diverse QA datasets with stopwords removed.

For evaluation, we assess InstEmb's performance on two categories of benchmarks: (1) **Instruction-following embedding benchmarks**, including Inst.STSb, IntentEmotion, NYTCluster Peng et al. (2024), and FollowIR benchmark(Weller et al., 2024a), which evaluate the model's ability to follow instructions and capture semantic similarity under various contexts; and InfoSearch benchmark(Zhou et al., 2024), which evaluates instruction-following capacities beyond content relevance. (2) **Generic sentence embedding tasks** from MTEB benchmark[1] including AskUbuntuDupQues, TwentyNewsgroups, SciDocsRR, and StackOverflowDup, which cover a wide range of semantic tasks for comprehensive embedding quality evaluation.

### 4.1.2 Evaluation Metrics

We employ Spearman correlation for Inst.STSb, the harmonic mean of success rates for IntentEmotion, mean average precision (mAP) for AskUbuntu, SciDocs, and StackOverflow, and V-measure for clustering tasks (NYTCluster and 20NewsGroups). **FollowIR** benchmark employs nDCG@5 for News21, MAP@1000 for Core17/Robust04, and p-MRR(Weller et al., 2024a) across all collections. normalized discounted cumulative gain at 5 (nDCG@5) measures ranking quality by cumulating the gains of retrieved documents, discounted logarithmically by their rank, and normalized by the ideal DCG. p-MRR metric ranges from –100 (indicating complete instruction contradiction) to 100 (reflecting perfect compliance). **InfoSearch** employs three metrics to evaluate performance: the Strict Instruction Compliance Ratio (SICR), the Weighted Instruction Sensitivity Evaluation (WISE)(Zhou et al., 2024), and p-MRR.

### 4.1.3 Baselines

We compare against three categories of models: (1) **General LLMs**: Llama2 (Touvron et al., 2023), Llama3 (Grattafiori et al., 2024), Mistral-7b-instruct (Jiang et al., 2023), DIFFEMBED (Zhang et al., 2025b), and PonTE(Yamada & Zhang, 2025) (2) **Fixed-instruction embedding models**: LLM2Vec (BehnamGhader et al., 2024), GritLM (Muennighoff et al., 2024), ECHO (Springer et al., 2024), and e5 (Wang et al., 2022); (3) **Instruction-adaptive models**: Instructor (Su et al., 2022), Inbedder (Peng et al., 2024), FollowIR (Weller et al., 2024a), Promptriever (Weller et al., 2024b).

Further implementation details and hyperparameters could be found in Appendix A.1.

## 4.2 Main Results Analysis

Table 1 and Table 2 present the primary evaluation results comparing our proposed InstEmb method against several baseline embedding methods across two categories of benchmarks: instruction-following embedding tasks and generic sentence embedding tasks.

**Instruction-Following Embedding Tasks.** Our InstEmb achieves new state-of-the-art performance across instruction-following benchmarks, demonstrating significant improvements in instruction comprehension. On the FollowIR benchmark (Table 1), our InstEmb achieves 28.5 average score and +15.6 p-MRR ratio - significantly surpassing previous SOTA methods like FollowIR-7B (24.8 score/+12.2 p-MRR) and Promptriever (26.1 score/+11.2 p-MRR). Remarkably, while these competitors use supervised training data specifically optimized for FollowIR, our model achieves strong performance through zero-shot inference without any benchmark-specific training. This result highlights InstEmb's strong generalization capability in instruction following retrieval.

On the InfoSearch benchmark, InstEmb$_{\text{MSE}}$ also demonstrates competitive performance, achieving a mean p-MRR of 7.7 with 20.1 SICR and 13.3 WISE scores. This represents a substantial improvement over the base Llama-3-8B-instruct model (6.6 p-MRR) and outperforms other strong baselines including FollowIR-7B (4.1 p-MRR). The consistent superiority across both FollowIR and InfoSearch

---

[1]The MTEB leaderboard can be found at `https://huggingface.co/spaces/mteb/leaderboard`

| Model | FollowIR | | | | | | | | InfoSearch | | |
| | Robust04 | | News21 | | Core17 | | Mean | | Mean | | |
| | MAP | p-MRR | nDCG | p-MRR | MAP | p-MRR | Score | p-MRR | SICR | WISE | p-MRR |
|---|---|---|---|---|---|---|---|---|---|---|---|
| GritLM-Reranker | 9.7 | +6.1 | 10.2 | +3.4 | 9.8 | +8.6 | 9.9 | +6.0 | 6.9 | -11.1 | -4.3 |
| Mistral-7B-instruct | 23.2 | +12.6 | 27.2 | +4.8 | 19.7 | +13.0 | 23.4 | +10.1 | 0.0 | -49.2 | -32.4 |
| DiffEMBED | 18.9 | +5.7 | 27.7 | +3.6 | 16.2 | +6.0 | 20.9 | +5.1 | - | - | - |
| FollowIR-7B | 24.8 | +13.7 | 29.6 | +6.3 | 20.0 | +16.5 | 24.8 | +12.2 | 12.5 | 13.4 | 4.1 |
| Promptriever | 28.3 | +11.7 | 28.5 | **+6.4** | 21.6 | +15.4 | 26.1 | +11.2 | - | - | - |
| Llama-2-7B-chat | 6.3 | +2.0 | 1.7 | +0.2 | 5.4 | +2.8 | 4.5 | +1.7 | 8.4 | -18.7 | -10.9 |
| InstEmb-MSE | 10.8 | +8.9 | 15.2 | +0.0 | 9.5 | +10.1 | 11.8 | +6.3 | 10.5 | -14.4 | -8.4 |
| Llama-3-8B-instruct | 12.7 | +2.3 | 17.6 | -1.0 | 11.7 | +1.7 | 14.0 | +1.0 | 19.6 | 13.1 | 6.6 |
| InstEmb-MSE | **29.2** | **+19.1** | **32.3** | +5.4 | **24.0** | **+22.4** | **28.5** | **+15.6** | **20.1** | **13.3** | **7.7** |

Table 1: Main results on FollowIR (left) and InfoSearch (right). InstEmb achieves state-of-the-art performance on FollowIR without extra supervised data tuned for this benchmark.

| Model | I.STSb | IntEmo. | NYT. | Mean | AskU. | 20news | SciD. | StackO. | Mean |
|---|---|---|---|---|---|---|---|---|---|
| LLM2Vec-llama2-7b | - | - | - | - | 63.13 | 51.04 | 84.03 | 51.02 | 62.30 |
| Echo-mistral-7b-eos | - | - | - | - | **64.1** | 53.04 | 83.68 | 51.84 | 63.16 |
| instructor-large | -15.02 | 47.96 | 49.96 | 27.63 | 63.48 | 53.51 | 81.83 | 50.50 | 62.33 |
| e5-large-v2 | 0.00 | 30.24 | 50.07 | 26.77 | 59.01 | 47.94 | 83.84 | 50.60 | 60.35 |
| llama-2-7b-chat$_{InputLast}$ | 23.24 | **95.56** | 56.87 | 58.55 | 55.70 | 47.43 | 76.58 | 41.78 | 55.37 |
| Inbedder | 22.07 | 89.68 | 64.65 | 58.80 | 60.32 | 52.33 | 80.61 | 44.77 | 59.51 |
| InstEmb-MSE$_{DAAP}$ | 23.51 | 93.45 | **70.77** | 62.57 | 60.16 | 52.56 | 80.43 | 47.37 | 60.13 |
| llama3-8b-instruct$_{InputLast}$ | 44.2 | 92.8 | 25.6 | 54.2 | 58.3 | 54.0 | 79.3 | 44.6 | 59.0 |
| PonTE | **44.60** | 89.50 | 48.03 | 60.71 | 57.49 | 52.71 | 82.47 | 41.97 | 58.66 |
| Inbedder | 23.1 | 94.5 | 62.0 | 59.9 | 61.9 | 54.8 | 83.2 | 46.2 | 61.52 |
| InstEmb-MSE$_{DAAP}$ | 41.37 | 94.12 | 65.76 | **67.08** | 63.25 | 54.26 | 84.86 | 48.40 | 62.69 |
| InstEmb-KL$_{DAAP}$ | 39.24 | 94.88 | 62.32 | 65.48 | 63.34 | **55.39** | **85.50** | **49.35** | **63.39** |

Table 2: Main results on Inst.STSb, IntentEmotion, NYTCluster, and all generic sentence embedding tasks. Subscripts "InputLast" and "DAAP" indicate the pooling method; Pooling methods' details can be found in the Appendix A.2. Inbedder under llama3-8b-instruct indicates our re-implementation of InbedderPeng et al. (2024).

benchmarks further validates InstEmb's robust instruction-following capabilities in diverse retrieval scenarios.

Beyond above, InstEmb$_{MSE-DAAP}$ attains a mean score of 67.08% on Inst.STSb, IntentEmotion, and NYTCluster. It gets an absolute improvement of 7.18% over the SFT baseline (59.9%). Consistent gains are also observed on llama2, where InstEmb$_{MSE-DAAP}$ improves the mean score by 3.77% over the SFT-based Inbedder (62.57% vs. 58.80%). In contrast to fixed-instruction embedding models (e5, instructor) that fail to model dynamic instructions effectively, InstEmb demonstrates robust advantages across all instruction-following embedding tasks.

**General Sentence Embedding Tasks.** Despite being optimized on instruction-following data, InstEmb demonstrates strong performance on general embedding benchmarks, achieving a competitive mean score of 63.39% (InstEmb-KL$_{DAAP}$) with 7× less training data than specialized baselines like LLM2Vec and Echo-mistral (both 1.5M examples). The performance of InstEmb on general sentence embedding tasks highlights the transferability and generalization ability of our instruction-based embedding method, an advantage also observed in its strong zero-shot performance on the FollowIR benchmark.

Experimental results demonstrate that MSE-based distillation achieves superior performance on instruction-following embedding tasks, while KL-divergence proves more effective for generic sentence embedding tasks. The following ablation study will provide a detailed analysis of this phenomenon.

# 5 ABLATION STUDIES & ANALYSIS

## 5.1 EFFECTIVENESS OF REPRESENTATION DISTILLATION

To validate the effectiveness of our representation distillation approach and understand the contribution of different loss functions, we conduct a controlled ablation study. Specifically, we remove $\mathcal{L}_{CL}$ and compare the performance of different training objectives applied to look-ahead tokens, allowing us to isolate the impact of distillation strategies on complementary semantic learning.

We evaluate two non-distillation methods and two distillation methods: **SFT** (standard autoregressive supervised fine-tuning without look-ahead tokens), **CE** (cross-entropy loss applied to look-ahead tokens without teacher guidance), **MSE** (eq 1), and **KL** (eq 2). The CE approach serves as a critical control, training look-ahead tokens using standard cross-entropy loss on target sequences rather than teacher-student distillation.

As shown in Table 3, several key observations emerge:

**Distillation Superiority**: Both MSE and KL distillation methods significantly outperform the CE baseline, demonstrating the effectiveness of teacher-guided semantic transfer. Confirming that distillation enables more effective complementary semantic learning than conventional token-level supervision.

**Task-Specific Specialization**: The two distillation objectives exhibit complementary strengths. KL divergence excels in generalizable tasks (higher Mean Gen.), likely due to its ability to preserve instruction-agnostic knowledge through probability distribution alignment. Conversely, MSE demonstrates superior performance on instruction-specific tasks (higher Mean Inst.), suggesting its effectiveness in fine-grained task nuance modeling through direct representation regression.

|  | Inst. | Gen. | Score | p-MRR |
|-----|-------|-------|-------|-------|
| SFT | 59.9 | 61.52 | 5.0 | +8.4 |
| CE | 60.13 | 61.93 | 27.3 | +14.8 |
| KL | 63.41 | 63.01 | 27.9 | +14.9 |
| MSE | 64.16 | 61.81 | 26.7 | +15.0 |

Table 3: Ablation study comparing different training objectives without contrastive learning.

## 5.2 IMPACT OF LOOK-AHEAD TOKEN LENGTH

we ablate the length of look-ahead tokens. During the inference phase, we employ different look-ahead lengths, specifically set to 0, 1, 4, and 8. When the look-ahead length is set to 0, it is equivalent to using only the last input token.

As shown in Fig. 2, it can be observed that almost all tasks exhibit a significant performance jump as the look-ahead token length increases from 0 to 1 Except several tasks such as IntSTS and intEmo, because these tasks were designed with very short inputs, resulting in high information density, the improvement brought by complementary semantics is not significant and may even decrease. However, when the look-ahead token length is further increased from 1 to 8, few tasks demonstrate scaling behavior. Based on these observations, we conclude that the presence of look-ahead is crucial, but the performance is not highly sensitive to the specific number of tokens added. Nevertheless, to accommodate certain tasks, such as NYTCluster, which show continuous and notable performance improvement with increasing look-ahead token length, we still recommend using a longer look-ahead sequence.

## 5.3 MULTI-VIEW CONTRASTIVE LEARNING ANALYSIS

Table 4 analyzes the contributions of different embedding views in the multi-view contrastive learning framework. Comparing the impact of removing the self-supervised dropout view (View 2; $\Delta$-5.78) versus removing both teacher input and student output views (Views 3&4; $\Delta$-1.75) highlights the critical role of the dropout-augmented view for contrastive stability, as its absence causes severe performance degradation and embedding collapse.

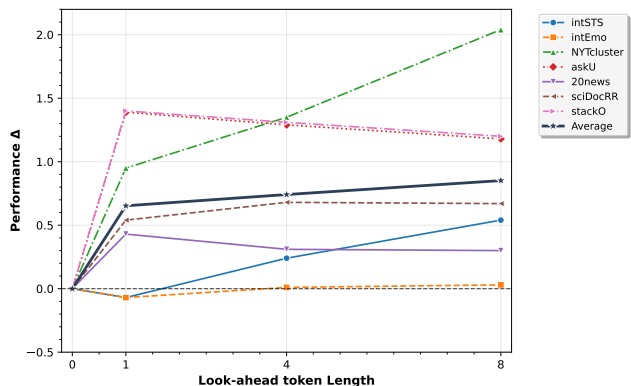

Figure 2: ablation study on Look-ahead token length

Retaining only student input and output views (-Views 2&3; Δ-2.96) demonstrates that a simple dual-tower contrastive setup between input and output embeddings also serves as a viable strategy.

Furthermore, the teacher input view (View 3), closely aligned with the distillation objective, provides essential complementary supervision; its removal (individually or jointly) consistently results in performance drops, underscoring its importance for overall embedding quality.

| Strategy | Inst. | Gen. |
|---|---|---|
| Full views | **67.08** | **62.69** |
| − view 2 | 61.30 $(-5.78)$ | 62.51 $(-0.18)$ |
| − view 3 | 65.05 $(-1.68)$ | 61.01 $(-2.03)$ |
| − view 4 | 65.54 $(-1.54)$ | 61.96 $(-0.73)$ |
| − view 2 & 3 | 64.12 $(-2.96)$ | 61.92 $(-0.77)$ |
| − view 2 & 4 | 56.44 $(-10.64)$ | 61.23 $(-1.46)$ |
| − view 3 & 4 | 65.33 $(-1.75)$ | 61.91 $(-0.78)$ |
| − All views | 64.16 $(-2.92)$ | 61.81 $(-0.88)$ |

Table 4: Ablation study on contrastive learning strategies. We remove views individually or jointly to measure their contributions. "−" means removing this view.

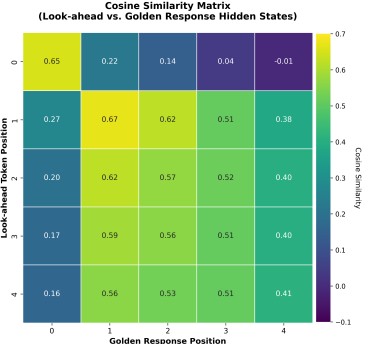

Figure 3: Cosine similarity matrices between the hidden states of the look-ahead token sequence and those of the golden output sequence.

## 5.4 IMPACT OF POOLING METHODS

We evaluate five pooling methods (SpecialFirst, SpecialMean, InputLast, AllMean, DAAP; details in Appendix A.2) on instruction-following embedding and generic sentence embedding tasks. As shown in Table 5, DAAP achieves the best overall performance. Our analysis reveals several important findings that highlight the effectiveness of different pooling strategies.

First, methods relying solely on special tokens (SpecialMean, SpecialFirst) perform worst on instruction understanding, indicating their limited discriminative power for capturing complex instructional semantics. Second, InputLast shows strong instruction understanding but trails DAAP in generic tasks, suggesting that input semantics alone are insufficient for optimal performance across all evaluation dimensions. Third, the undifferentiated fusion approach of AllMean, which treats all tokens equally, leads to degraded performance compared to DAAP, demonstrating the importance of strategic token selection and weighted integration in embedding extraction.

| Pooling Method | Inst. | Gen. |
|---|---|---|
| SpecM | $55.22(-11.86)$ | $59.10(-3.59)$ |
| SpecF | $62.69(-4.39)$ | $61.76(-0.93)$ |
| InputL | $66.72(-0.36)$ | $61.80(-0.89)$ |
| InputL+SpecF | $66.51(-0.56)$ | $61.51(-1.14)$ |
| AllMean | $63.48(-3.60)$ | $61.63(-1.03)$ |
| **DAAP** | **67.08** | **62.69** |

Table 5: Performance comparison of different pooling strategies. SpecM: SpecialMean, InputL: InputLast, SpecF: SpecialFirst.

| | Inst. | Gen. | Score | p-MRR |
|---|---|---|---|---|
| $InstEmb_{absQA}$ | 67.08 | 62.69 | 28.5 | +15.6 |
| $InstEmb_{extQA}$ | 66.51 | 62.39 | 27.3 | +14.8 |
| $InstEmb_{MS-MARCO}$ | 66.56 | 62.92 | 29.4 | +15.7 |

Table 6: Performance across different training datasets, demonstrating consistent robustness.

## 5.5 SEMANTIC ALIGNMENT VISUALIZATION OF LOOK-AHEAD TOKENS

To understand the semantic role of look-ahead tokens, we compute cosine similarities between the hidden states of: (1) the last input token and subsequent look-ahead tokens of InstEmb, and (2) the last input token and subsequent golden output tokens of LLaMA3.

As shown in Figure 3, we observe two key patterns: (1) the last input token embedding shows low similarity with other positions, indicating its focus on capturing primary input semantics; (2) look-ahead tokens exhibit consistently higher similarity with the golden output sequence, demonstrating their role in capturing complementary semantics aligned with the intended output. Extra attention pattern visualization could be found in Appendix A.3

## 5.6 ROBUSTNESS ACROSS DIVERSE TRAINING DATASETS

To assess model robustness across different training datasets, we conduct experiments using two alternative datasets, both trained with the MSE objective.

The first dataset is an **Extractive QA Dataset (extQA)** containing approximately 150k examples from SQuAD_v2 (Rajpurkar et al., 2018) and NewsQA (Trischler et al., 2016), where answers are directly extracted from given contexts.

The second dataset is **MS-MARCO** (Nguyen et al., 2016), comprising around 80k examples characterized by longer answer sequences and more diverse, free-form generation patterns compared to extractive QA.

Table 6 presents the performance results. We observe minimal performance variation across different training datasets, indicating that our method maintains robust performance regardless of the underlying data paradigm. This stability enables straightforward utilization of diverse open-source datasets without requiring dataset-specific normalization or complex preprocessing.

## 6 CONCLUSION

We introduce **InstEmb**, a novel instruction following embedding framework that effectively captures both primary and complementary semantic information through representation distillation and supervised contrastive learning. By leveraging learnable look-ahead tokens, InstEmb encodes rich instruction-conditioned semantics directly within a single prefilling step, significantly boosting embedding performance without additional inference latency. Experiments demonstrate that InstEmb achieves state-of-the-art results across multiple instruction following benchmarks. Our work provides an efficient yet powerful solution for instruction following embedding generation, paving the way for future research towards more effective and computationally efficient embedding methods.

## REPRODUCIBILITY STATEMENT

For reproducibility: methodological details are in Subsection 3; dataset descriptions (all publicly available) in Subsection 4.1.1; implementation details in Appendix A.1. Core logic code is provided in supplementary materials.

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

# A APPENDIX

## A.1 IMPLEMENTATION DETAILS

We employ LLaMA-3-8B-Instruct as the backbone model. During knowledge distillation, we freeze the first 24 layers of the student model and all teacher model layers. InstEmb is trained for 1 epoch using Adam optimizer with learning rate $5 \times 10^{-6}$. The sequence length for look-ahead tokens is set to 8, which depends on data distribution. For ablation studies on training data, we set the look-ahead token length to 32 for MS-MARCO due to its longer average output length. However, during inference, we uniformly use a look-ahead length of 8 for all datasets to ensure consistency and fair comparison. For contrastive learning, we use dropout ratio 0.2 for view2 and temperature $\tau = 0.1$. Training uses maximum input sequence length 512, batch size 8, gradient accumulation steps 6, and bf16 precision on 8 NVIDIA H800 GPUs.

## A.2 POOLING METHOD DETAILS

- **SpecialFirst**: Selects the hidden state of the first special token as the representation:

$$e = h^S_{|\boldsymbol{x}|+1} \tag{6}$$

- **SpecialMean**: Computes the mean embedding over all special tokens:

$$e = \frac{1}{L} \sum_{j=1}^{L} h^S_{|\boldsymbol{x}|+j} \tag{7}$$

- **InputLast**: Utilizes the hidden state of the last input token:

$$e = h^S_{|\boldsymbol{x}|} \tag{8}$$

- **AllMean**: Computes the mean embedding over the last input token and all special tokens:

$$e = \frac{1}{L+1} \left( h^S_{|\boldsymbol{x}|} + \sum_{j=1}^{L} h^S_{|\boldsymbol{x}|+j} \right) \tag{9}$$

- **DAAP**: Our proposed method explicitly integrates input and output semantics by averaging the last input token embedding (capturing input semantics) and the mean embedding of special tokens (capturing output semantics):

$$e = \frac{1}{2} \left( h^S_{|\boldsymbol{x}|} + \frac{1}{L} \sum_{j=1}^{L} h^S_{|\boldsymbol{x}|+j} \right) \tag{10}$$

## A.3 ATTENTION PATTERN VISUALIZATION

As shown in Figure 4, we further conducted additional attention pattern visualizations and observed notable differences. Specifically, Llama3-8b-instruct exhibits a prominent *attention sink* (Xiao et al., 2023) effect, where a substantial amount of attention fails to focus on meaningful information and instead defaults to the first position.

In contrast, after training, our InstEmb demonstrates a more sophisticated attention allocation strategy. Rather than concentrating attention indiscriminately on the first token due to uncertainty about relevant information, InstEmb selectively focuses on several semantically critical positions. These key positions include the end of the system prompt, which serves as a global instruction, and the end of the instruction. This selective attention pattern indicates that InstEmb has developed enhanced sensitivity to fine-grained instruction semantics, enabling more effective modeling of task-specific representations through dynamic association with instruction-relevant content.

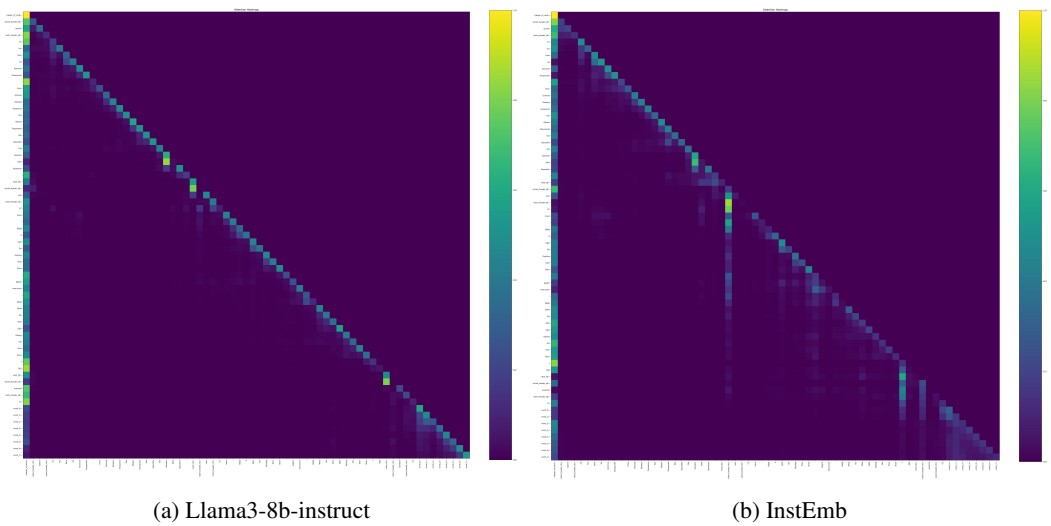

(a) Llama3-8b-instruct          (b) InstEmb

Figure 4: Comparison of attention patterns.

## A.4 INSTRUCTION-ROBUSTNESS EVALUATION

Following Peng et al. (2024), we measure the embedding model's robustness to instruction perturbations. Let $\Delta_{ci}$ denote the average-score gap between *correct* and *incorrect* instructions, and $\Delta_{ii}$ the gap between *implicit* and *incorrect* ones. Larger deltas indicate stronger robustness.

| Model | $\Delta_{ci}$ | $\Delta_{ii}$ |
|---|---|---|
| Instructor-Large | 0.02 | 0.01 |
| Llama2-7B-Chat | 0.19 | 0.17 |
| Inbedder | 0.21 | 0.18 |
| **InstEmb** | **0.26** | **0.26** |

Table 7: Instruction-robustness on **FewNerd** tasks. We employ InstEmb based on Llama2-7b-chat.

InstEmb achieves the largest margins and $\Delta_{ci} \approx \Delta_{ii}$, confirming that it retains task understanding even under implicit wording.

# B STATEMENT

## B.1 LLM USAGE

In this work, Large Language Models (LLMs) are employed in a supplementary capacity to enhance the research process rather than as primary content generators. Specifically, LLMs are utilized for language polishing and straightforward content expansion to improve the clarity and readability of the manuscript. Additionally, Internet-enabled LLM agents serve as a extra supplementary research tool to help retrieve and identify relevant related work from the existing literature. It is important to note that LLMs are not used as complete chapter authors nor as creators of the overall thesis framework, ensuring that the core intellectual contributions and structural design remain the original work of the author.

## B.2 ETHICS STATEMENT

### DATA PROVENANCE AND USAGE

All datasets used in this work are publicly available benchmark datasets obtained from legitimate academic sources. Data acquisition and usage comply with original licenses and terms of use. No sensitive or personal identifiable information was involved in this research.

### RESEARCH INTEGRITY

This work upholds high standards of scientific excellence through transparent methodology and reproducible experiments. The research represents an honest advancement in *[representation learning]* without misrepresentation of results.

### SPONSORSHIP DECLARATION

Computational resources and experimental conditions were provided by JD.com. The sponsor had no role in study design, data analysis, or interpretation of results. No conflicts of interest exist.

### SOCIETAL CONSIDERATIONS

As a technical contribution focused on algorithmic improvement, this work poses minimal ethical risks. We encourage responsible application of the proposed methods and will release code to promote reproducibility and broader scientific benefit.

