# OpenReview forum: "InstEmb: Instruction-Following Embeddings through Look-Ahead Token Distillation"
_ICLR.cc/2026/Conference — Submitted to ICLR 2026_

### Official Review · Reviewer_GAWy · 2025-10-26

**Soundness:** 3
**Presentation:** 2
**Contribution:** 3
**Rating:** 6
**Confidence:** 2

**Summary:**

This paper introduces and assesses a new method, InstEmb, for aligning instruction following models, using teacher-based latent representations. InstEmb uses both cumulative (last token) and granular single-token representations to capture semantic information without additional decoding steps from the teacher models.

**Strengths:**

- Well scoped methodology and contribution that combines existing methods for token alignment and soft-prompting to improve instruction tuning
- Offers clear benefits of efficiency and performance improvements on the QA datsets, and evidence for generalization across new datasets

**Weaknesses:**

Most of my concerns are around the presentation and claims of semantic information types:

- Grounding of "primary" and "complementary" semantics claims
    - It's unclear how these notions are operationalized or captured in the methodology -- there are no clear experiments of qualitative assessments that these correspond to distinct, measurable factors in the latent space;  and,
    - The work states these two are "jointly" optimized but doesn't fully quantify what information is captured by each representation (e.g., L450 shows that the lookup tokens align with the answer tokens; but this to me is a bit obvious given the alignment objective)

- The figures are difficult to read (see suggestions in the Questions section), and the scoring methods are unclear. I see that the authors' method excel on these metrics, but it is unclear to me what these metrics capture and how this supports the conclusion of better aligned semantics.
    - E.g., missing explanation of nDCG@5

**Questions:**

1. What is the overlap (i.e., Fig. 2) of the look-ahead token with the last token? With the original instruction?
2. What is the benefit of using the same model as the student and teacher? Is there a performance difference when using a stronger teacher model to distill?
3. (Suggestion) A qualitative case study on which instructions perhaps benefit the most from the "semantic" information captured with this methodology.

Below are presentation suggestions and references to typos:
- L54: "sampling during decode" --> decoding?
- L260: "strongity" --> strength?
- L278: Please elaborate on "nDCG@5" in this section
- Fig. 1: The caption does not explain "view1/view2", font size on text is small and difficult to read
- Fig. 2: The values and titles are way too small to read, please increase the font size

---

> ### Author Response · Authors · 2025-11-20
>
> We are deeply grateful for your thorough and insightful review of our work. Your thoughtful comments and constructive questions have been invaluable in helping us strengthen our work and clarify its presentation. Below, we provide a point-by-point response to the specific weaknesses and questions you raised.
>
> ## Q1
> As described in new Section 5.5, Figure 3 (original Sec.5.4 Figure.2) is used to observe how the lookahead tokens align with the semantic space of the actual generated tokens, and how the hidden state distributions of the lookahead tokens distinguish the input last token. The matrix in the figure has dimensions of (1+n, 1+n), where 1 represents the input last token and n denotes the number of look-ahead tokens. The figure shows that the semantic similarity between the lookahead tokens and the input last token is relatively low, while the semantic similarity between the lookahead tokens and the actual generated tokens is relatively high. This also indirectly suggests that the semantic spaces of these two types of tokens are distinct.
>
> ## Q2
> This is an excellent question that has frequently come up in our internal discussions as well.
>
> **Original Intent of our Method**:
> Our approach is **not** traditional knowledge distillation from a large model to a small one. Instead, our goal is to enable the model to align with its own future autoregressive representations without introducing additional decoding overhead. This objective is twofold:
> 1. We avoid multi-step decoding during inference.
> 2. We align representations in the same latent space.
>
> **Teacher Model Choice**:
> To achieve these goals, we must use a teacher model that is isomorphic with the student model.  Only under this condition can the representation spaces be consistent, allowing the student to "preview" and align with its own future states via distillation loss. Using a larger or architecturally different model would disrupt this self-preview mechanism and undermine the intended effect.
>
> ## To Weakness & Suggestion:
> The primary weakness pertains to the semantics of both the primary and complementary components. These two types of semantics essentially correspond to instruction + input and output, respectively. We have revised Lines 43–51 in the original manuscript to explicitly clarify these definitions. Once these concepts are clearly defined, how these semantics are captured by the training objective becomes straightforward. We align the input's last token and the look-ahead token of the student model—representing **instruction + input** and **output**, respectively—with the corresponding positions in its isomorphic teacher model. Additionally, we incorporate contrastive learning on the input's last token to further enhance the primary semantics. The visualization in Figure 3 already addresses this question, although it may appear relatively straightforward given the alignment objective.
>
> In our current model version, the look-ahead component is unable to generate fluent text, as it was not trained for autoregressive tasks but rather for representation alignment. From this perspective, we still find it challenging to present clear case studies.
>
> ## Presentation Revision
> Based on your feedback regarding the presentation, we have:
> - Corrected wording errors throughout the document.
> - Improved the readability of Figure 1 and the original Figure 2 (now Figure 3).
> - Added an explanation of the nDCG metric, along with a citation for it, so that readers can easily find its details.
>
>
> Hoping I have correctly understood all your questions and my responses and revisions have adequately addressed your concerns.  If not, please feel free to ask me, and I will do my best to answer them.

---

> > ### Comment · Reviewer_GAWy · 2025-11-25
> >
> > Thank you for the response and clarifications, I will maintain my assessment of the work.

---

### Official Review · Reviewer_BmMX · 2025-10-30

**Soundness:** 2
**Presentation:** 2
**Contribution:** 2
**Rating:** 4
**Confidence:** 4

**Summary:**

This paper proposes InstEmb, a new instruction-following embedding framework that captures both primary and complementary semantics through representation distillation and supervised contrastive learning. The method introduces learnable look-ahead tokens to distill output-related semantic signals. This paper also introduces a new pooling strategy to explicitly combine both semantic signals.

**Strengths:**

1. The paper addresses the critical and timely problem of building fine-grained, instruction-adaptive embeddings for modern retrieval systems.

2. The core technical approach is sound. The use of learnable look-ahead tokens coupled with representation distillation offers an elegant and efficient solution.

**Weaknesses:**

1. **Incremental Conceptual Novelty:**
While the proposed framework is neatly integrated, its constituent components—knowledge distillation, contrastive learning, and soft prompt-like tokens—are all well-established techniques. The contribution of InstEmb lies primarily in their combination rather than in introducing fundamentally new algorithmic or theoretical insights. As a result, the conceptual novelty may be viewed as incremental rather than groundbreaking.

2. **Unclear Definition of Core Concepts:**
The key notions of primary semantics and complementary semantics are introduced in Section 1 without adequate clarification. This lack of explicit definition may confuse readers, as the distinction underpins much of the subsequent methodology. It appears that primary semantics correspond to the semantics derived from the input and instruction, while complementary semantics relate to the output or response information. If this interpretation is inaccurate, the authors should clarify the terminology and provide an intuitive explanation early in the paper.

3. **Insufficient Justification for the Joint-Usage Hypothesis:**
The central claim that “both primary and complementary semantics need to be used simultaneously” is not sufficiently supported. Prior work, such as InBedder, already demonstrates the benefit of modeling complementary (output-related) semantics over primary-only embeddings (e.g., Instructor). However, the paper does not provide clear empirical evidence that combining both types yields synergistic improvements beyond using complementary semantics alone. A dedicated ablation study or comparative experiment would be necessary to substantiate this key hypothesis.

4. **Missing Comparison with a Highly Relevant Baseline:**
The related work and experimental sections overlook an important recent study, “*Don’t Reinvent the Wheel: Efficient Instruction-Following Text Embedding based on Guided Space Transformation*” (ACL 2025), which also targets instruction-aware embedding optimization. Since that work addresses a conceptually similar problem using a geometric transformation approach, a comparison—either qualitative or empirical—would help position InstEmb more precisely within the current research landscape and highlight its distinct contributions.

**Questions:**

1. Given that InstEmb primarily combines known techniques (knowledge distillation, contrastive learning, and soft prompt tuning), what aspects of the framework should be considered conceptually new beyond this integration?

2. Could the authors provide a clearer and more formal definition of primary semantics and complementary semantics, ideally with illustrative examples, to help readers understand how these two components differ and interact?

3. Are there specific ablation studies or quantitative results demonstrating that using both primary and complementary semantics jointly leads to better performance than using either alone?

---

> ### Author Response · Authors · 2025-11-20
> **comment(1/2)**
>
> Thank you for your thoughtful and constructive feedback. We sincerely appreciate your insightful comments, which have helped us further clarify our methodological contributions and experimental design. Below, we provide a point-by-point response to the issues you raised. We have noted your focus on the novelty of our work and hope our responses address your concerns.
>
> ## W1 & Q1
> Our core innovation lies in addressing two critical challenges faced by embedding models in real-world applications through a novel problem formulation and solution:
>
> **Practical Challenges & optimization objective**
>    Embedding models in practice face two key issues:
>    - They cannot afford the computational overhead of multi-step decoding due to massive invocation scales.
>    - Relying solely on a single positional encoding (e.g., the last token) fails to capture the full information of the output sequence.
>    This led us to formulate a new core problem: How can a model "preview" its own future hidden states without performing multi-step decoding?
>
> **Technology selection based on optimization objectives**
>    Our design is strictly aligned with the core problem:
> - We use a frozen, isomorphic teacher model to constrain optimization within the student’s own parameter space. This is not traditional knowledge distillation from a large to a small model, but rather a mechanism for the student to align with its own future output representations via distillation loss.
> - In our work, Look-ahead tokens are designed not as task-specific adapters(like soft prompt) but as semantic carriers of future outputs. They allow the model to capture the semantics of future output tokens in a single forward pass, avoiding both the inefficiency of re-encoding and the discreteness of token-level autoregression.
>
> In summary, while the task forms may resemble existing methods, the technical composition is entirely driven by our novel objective. The training techniques are supportive in nature, not the core contribution.
>
> **In terms of the timeliness and popularity of the research**, recent work in speculative decoding has explored the use of look-ahead tokens for parallel decoding (like <https://arxiv.org/pdf/2504.18583>  and <https://arxiv.org/pdf/2410.05589>), to the best of our knowledge, the approach of enhancing the instruction following ability of embedding using look-ahead tokens represents the early exploration within the context of LLM-based representation method. It is worth noting that concurrent submission at ICLR 2026 (<https://openreview.net/forum?id=okjogxO1Fu> or its arxiv version https://www.arxiv.org/pdf/2509.24291) has also begun to investigate a similar mechanism, which further validates the relevance and timeliness of our problem formulation.
>
>
> ## W2 & Q2
>
> You are absolutely right. The two types of semantics indeed refer to **instruction + input** and **output**, respectively. We have revised Lines 43–51 in the original paper to explicitly clarify these definitions. Thank you for pointing this out.
> >... The last token primarily captures the semantics of the input and instruction but fails to incorporate the semantic information of output, as the latter is distributed across multiple critical tokens and cannot be adequately aggregated into a single position. To clarify this distinction, we hereafter refer to these two roles as **primary semantics** and **complementary semantics.**

---

> ### Author Response · Authors · 2025-11-26
> **comment(2/2)**
>
> ## W3 & Q3
>
> From the existing experiment, it can be inferred that there is a necessity for the simultaneous use of two types of semantics.
>
> First, Inbedder's **first-generation** pooling was their strongest method. **average generation** pooling over full output tokens was inferior to first-gen pooling and incurred additional latency. We guess this is due to its SFT training objectives.
>
> In our work, the **InputL** method in Table 5 corresponds to Inbedder's first-generation pooling. InputL achieves competitive performance, and we do not dispute its effectiveness. Furthermore, the **DAAP** result in Table 5 outperforms both InputL and SpecM, demonstrating that jointly leveraging both types of semantics is beneficial. Our work builds upon Inbedder by proposing a more effective way to incorporate output-related semantics, which is also aligned with our training objective.
>
> In addition to the above, our revised paper adds a **new Section 5.2** with an ablation study on look-ahead token length. The main conclusion is that introducing even a single look-ahead token (length=1) leads to a performance jump across most tasks. This strongly supports the advantage of incorporating complementary semantic information. **A similar conclusion could be obtained in W3&Q3 of reviewer xd92**
>
> ## W4
>
> Thank you for recommending this baseline. We were already aware of this creative and highly efficient method. However, we encountered challenges in aligning its evaluation with existing works:
> - In the GSTransformer paper, the authors also use Inbedder as a baseline. However, the reported Inbedder score on NYTCluster is 72.70, whereas the original Inbedder paper reports 64.65—a gap of ~8 points. A similar discrepancy occurs on IntEmo.
>
> Due to these inconsistencies, we decided not to include GSTransformer as a direct baseline. However, we fully acknowledge its relevance and have **added it to the Related Work section** in the revised paper
>
> Thank you once again for your valuable time and feedback. Hoping these revisions substantially improve the clarity, rigor, and completeness of our work.

---

### Official Review · Reviewer_xd92 · 2025-10-31

**Soundness:** 3
**Presentation:** 3
**Contribution:** 3
**Rating:** 4
**Confidence:** 4

**Summary:**

This paper proposes InstEmb, an instruction-following embedding framework designed to enhance semantic adaptability in text embeddings derived from large language models (LLMs). The framework jointly optimize primary semantics via contrastive learning focused on the final input token and complementary semantics via representation distillation from a frozen teacher model, using learnable look-ahead tokens that emulate future output semantics without incurring decoding latency. The framework also fuses the last-input-token and look-ahead-token representations to better align training objectives with inference-time embeddings.

Expriments on instruction-following benchmarks (FollowIR, Inst.STSb, IntentEmotion, NYTCluster) and generic sentence embedding tasks (MTEB subset) show that InstEmb achieves strong results compared to baselines such as InBedder, FollowIR, and Promptriever. The paper also includes ablations on distillation methods, contrastive views, pooling strategies, and training datasets.

**Strengths:**

1.	The paper proposes an efficient framework that trains an instruction-following embedding model in a single LLM pass, offering a clean and practical solution.
2.	The experimental section is extensive, covering both instruction-following and generic embedding benchmarks with strong baselines. The ablation studies provide insights into design choices.
3.	The motivation to bridge the gap between instruction-following generation and embedding representation is well articulated and timely. The authors clearly identify two limitations of existing embedding methods and directly address them through an well-designed framework.

**Weaknesses:**

1.	The current ablation studies on contrastive learning (§5.2) are conducted on top of the representation distillation module, without providing clear comparisons between the two modules individually. A proper factorial ablation (e.g., SFT + Distillation vs. SFT + Contrastive) would better reveal which component contributes more to the overall gain.

2.	Teacher–student configuration not sufficiently explored. As described in Appendix A.1, both the teacher and one student model share the same base architecture (LLaMA-3-8B-Instruct). No experiments are conducted with a larger or more capable teacher model, making it unclear whether the observed improvements stem from the proposed framework or from the inherent capacity of the backbone model.

3.	The number of look-ahead tokens is fixed to 8 throughout all experiments, without any sensitivity or scaling study. It is therefore uncertain how this hyperparameter affects model performance, training stability, or inference efficiency.

4.	Section 5.5 mentions training with the MS-MARCO dataset, which lacks explicit instruction fields. It remains ambiguous how instruction semantics are preserved — is the question field treated directly as the instruction?

5.	Some related works are missing which should be discussed, used as baseline or evaluation [1-7].

[1] Oh, Hanseok, et al. "Instructir: A benchmark for instruction following of information retrieval models." arXiv preprint arXiv:2402.14334 (2024).

[2] Yoo, Young Hyun, et al. "Hyper-CL: Conditioning Sentence Representations with Hypernetworks." Proceedings of the Annual Meeting of the Association for Computational Linguistics. Vol. 1. Association for Computational Linguistics (ACL), 2024.

[3] Sun, Weiwei, et al. "MAIR: A Massive Benchmark for Evaluating Instructed Retrieval." Proceedings of the 2024 Conference on Empirical Methods in Natural Language Processing. 2024.

[4] Zhou, Jianqun, et al. "Beyond Content Relevance: Evaluating Instruction Following in Retrieval Models." The Thirteenth International Conference on Learning Representations.

[5] Feng, Yingchaojie, et al. "Don't Reinvent the Wheel: Efficient Instruction-Following Text Embedding based on Guided Space Transformation." arXiv preprint arXiv:2505.24754 (2025).

[6] Yamada, Kosuke, and Peinan Zhang. "Out-of-the-Box Conditional Text Embeddings from Large Language Models." arXiv preprint arXiv:2504.16411 (2025).

[7] Zhang, Gaifan, Yi Zhou, and Danushka Bollegala. "CASE--Condition-Aware Sentence Embeddings for Conditional Semantic Textual Similarity Measurement." arXiv preprint arXiv:2503.17279 (2025).

**Questions:**

1.	Could you provide results comparing SFT + Distillation and SFT + Contrastive training separately, to determine which module contributes most to the improvements?

2.	Why was the teacher model kept similar in scale to the student? Have you tried a larger teacher, or could you comment on how performance might scale with teacher capacity?

3.	Have you explored varying the number of look-ahead tokens (e.g., 4, 8, 16)? Does the performance plateau or degrade as this number changes?

4.	Regarding the MS-MARCO experiments (§5.5), since the dataset lacks explicit instructions, do you treat the query text as the instruction itself? If so, how can the model still learn genuine instruction-following behavior rather than mere query encoding?

5.	Would combining datasets with and without explicit instruction fields affect the generalization ability of InstEmb?

---

> ### Author Response · Authors · 2025-11-20
>
> We sincerely appreciate your insightful comments, which have helped us identify areas where our explanations and experiments could be improved. Below, we provide a point-by-point response to the issues you raised.
>
> ## W1 & Q1
> Thank you for pointing out the need for clearer distinctions between SFT and representation distillation. In our work, SFT and representation distillation are conceptually parallel rather than sequentially stacked losses. We believe your underlying question pertains to the individual contributions of **look-ahead token usage**, **representation distillation**, and **contrastive learning**.
> Existing ablation studies can address this:
> - **Ablation on Look-ahead Token**:
>   As shown in Table 3, we compare the **SFT** (using ground-truth `input_ids` to predict `labels`) with the **CE** (using look-ahead tokens to predict `labels`). Both settings use the cross-entropy loss and do not involve a teacher model. The introduction of look-ahead tokens leads to comparable or slightly improved performance on instruction and general tasks, with a notable improvement on the FollowIR dataset and there is a slight improvement in other tasks.
> - **Ablation on Representation Distillation**:
>  Also in Table 3, we compare the **CE** setting (without teacher) with **MSE** or **KL** settings (with teacher). Here, the teacher receives the full input and output, while the student receives the input and look-ahead tokens. The loss is replaced with MSE or KL divergence. We observe a significant performance gain on instruction-related tasks (from 60.13 to 64.16).
> - **Ablation on Contrastive Learning**:
>   By comparing the **MSE** results in Table 3 and the **InstEmb-MSE_(DAAP)** results in Table 2, we note a further improvement (64.16 → 67.08). However, the gain is less pronounced than that from representation distillation.
>
> Hoping these results clarify the contributions of each component.
>
> ## W2 & Q2
> This is an excellent point that has frequently come up in our internal discussions as well.
>
> **Original Intent of our Method**:
> Our approach is **not** traditional knowledge distillation from a large model to a small one. Instead, our goal is to enable the model to align with its own future autoregressive representations without introducing additional decoding overhead. This objective is twofold:
> 1. We avoid multi-step decoding during inference.
> 2. We align representations in the same latent space.
>
> **Teacher Model Choice**:
> To achieve these goals, we must use a teacher model that is isomorphic with the student model.  Only under this condition can the representation spaces be consistent, allowing the student to "preview" and align with its own future states via distillation loss. Using a larger or isomerous model would disrupt this self-preview mechanism and undermine the intended effect.
>
> ## W3 & Q3
> You raised a valid and important point regarding the investigation into the effect of look-ahead token length. We agree that this was a necessary experiment to include, and we have now conducted it as suggested.
>
> | dataset | L=0 | L=1 (Δ) | L=4 (Δ) | L=8 (Δ) |
> |--------|------------|---------|---------|---------|
> | **intSTS** | 41.30 | -0.07 | +0.24 | +0.54 |
> | **intEmo** | 94.24 | -0.07 | +0.01 | +0.03 |
> | **NYTcluster** | 67.62 | +0.95 | +1.35 | +2.04 |
> | **askU** | 61.78 | +1.39 | +1.29 | +1.18 |
> | **20news** | 53.84 | +0.43 | +0.31 | +0.30 |
> | **sciDocRR** | 84.18 | +0.54 | +0.68 | +0.67 |
> | **stackO** | 46.93 | +1.40 | +1.31 | +1.20 |
> | **Average** | - | +0.65 | +0.74 | +0.85 |
>
> **Look-ahead token effectiveness lies in presence, not quantity**
> It can be observed that almost all tasks exhibit a significant performance jump as the look-ahead token length increases
> from 0 to 1.  Except several tasks such as IntSTS and intEmo, because these tasks were designed with very short
> inputs, resulting in high information density, the improvement brought by complementary semantics
> is not significant and may even decrease.
>
> **performance plateau**
> We conclude that the presence of look-ahead is crucial, but the performance is not highly sensitive to the specific number of tokens
> added. Nevertheless, to accommodate certain tasks such as NYTCluster, which show continuous and notable performance improvement with increasing look-ahead token length, we still recommend using a longer look-ahead sequence.
>
> The key findings and analysis are also presented in the new Section 5.2 of our revised paper.

---

> ### Author Response · Authors · 2025-11-20
>
> ## W4 & Q4 & Q5
> This concern is about the conflation of **query text** and **instruction**. Indeed, in an ideal setting, instructions should be treated as meta-information, separate from user-provided content. In the absence of an instruction field, we note that some related works have concatenated fixed instructions:
>
> - **Promptriever** states:
>   "when using the MS MARCO dataset, the standard ‘instruction’ is ‘Given a web search query, retrieve relevant passages that answer the query’ which is prepended to every query."
> - **FollowIR** does similar.
>
> In practice, we use query as our instruction and incorporate a fixed system instruction during training, which provides a global, abstract instruction independent of the query.
>
>
> ## W5
> We thank the reviewer for the valuable suggestions regarding relevant baselines and benchmarks. In response, we have add extra experiments in our revised paper as follows:
>
> - We have included the benchmark from **[4] Zhou et al., ICLR 2025** in **Table 1**.
> - We have added baseline results from **[6] Yamada & Zhang, arXiv:2504.16411** to **Table 2**.
> - We have incorporated discussions of **[2] Yoo et al., ACL 2024**, **[5] Feng et al., arXiv:2505.24754**, and **[7] Zhang et al., arXiv:2503.17279** into the **Related Work** section.
>
> We note, however, that the results reported in **[5]** for **Inbedder** on NYTCluster is 72.70 appear inconsistent with those in the original work (64.65 in the original Inbedder paper, and IntEmo as well) . Given these differences in evaluation, we have refrained from direct comparisons with **[5]**.
>
> Regarding the other suggested works, due to time constraints and computational costs, we are unable to provide results at this time. We will make our best efforts to include most of the remaining baselines and benchmarks in the final version.
>
>
> Thank you once again for your valuable time and feedback. We believe these revisions substantially improve the clarity, rigor, and completeness of our work.

---

> > ### Comment · Reviewer_xd92 · 2025-11-24
> >
> > Thank you for your replies. I still have one question regarding W4. Since you use the query as the instruction, would this be conceptually different from other formulations of instruction-following—for example, instructions that focus on different aspects of a document or that define relevance in other ways? In other words, would a model trained with the query-as-instruction paradigm generalize effectively to other instruction-following scenarios?

---

> > > ### Author Response · Authors · 2025-11-25
> > >
> > > Thank you for your thoughtful follow-up question. We appreciate the opportunity to further clarify this point. To this question, our answer is that our model can generalize to other instruction-following scenarios. And we believe it is necessary to give a quantitative analysis of it.
> > >
> > > Following §3.3 “Instruction Robustness Tests” in *Inbedder* (Answer is All You Need: Instruction-following Text Embedding via Answering the Question), we conduct the same experiment, **which evaluates instruction robustness by testing with generalized versions of the instruction**.
> > >
> > > .  Experiment definition is as follows:
> > > > This task evaluates the model's performance under correct, implicit, and incorrect instructions through clustering tasks. For each clustering task, GPT-4 is first guided to paraphrase the original task instructions, generating 10 correct instructions. Simultaneously, GPT-4 is used to rephrase the instructions into implicit expressions, yielding 10 implicit instructions. Additionally, GPT-4 is employed to generate 10 incorrect instructions that deviate from the original task objective. **All these datasets are clustered under a complex task instruction, such as entity type, the aspect of the review, or the reason to (dis)like**. During evaluation, the performance difference between the average performance of correct instructions and incorrect instructions is used as **Δci**, while the performance difference between the average performance of implicit instructions and incorrect instructions is used as **Δii**. These metrics measure the model's robustness, with larger delta values indicating stronger instruction robustness.
> > >
> > > To stay consistent with Figure 5 in Inbedder, we run the evaluation on the **Llama2-7b-chat** based **InstEmb** model.
> > >
> > > | Model                   | Δci  | Δii  |
> > > |-------------------------|------|------|
> > > | Instructor-Large        | 0.02 | 0.01 |
> > > | Llama2-7B-Chat          | 0.19 | 0.17 |
> > > | Inbedder                | 0.21 | 0.18 |
> > > | **InstEmb**             | **0.26** | **0.26** |
> > >
> > > InstEmb obtains the largest margin and, notably, Δci is similar to Δii, showing that the model still understands the task even when the instruction is turned implicit.
> > >
> > > In addition to the quantitative experiments mentioned above, we thought that many related works employ hard mining techniques to construct training data with different instructions for the same input at the data level.  In our approach, we utilize the MS-MARCO dataset, where the same passage is typically associated with multiple semantically diverse queries and corresponding outputs for each query.
> > > Functionally, these passages and related queries are equivalent to the paradigm of "issuing different task instruction for the same context." During the training phase, we directly concatenate the query with the passage in the following format: `Input: {passage}\n\nInstruction: {query} → {answer}` .Since the answer spaces for different queries pertaining to the same passage are almost non-overlapping, the model must learn to "extract or generate specific types of information based on the instruction (query)." This aligns with the paradigm of explicit instruction, where "the same input + different instructions → different outputs.
> > >
> > > What's more, our model's ability to generalize stems from its training objective. We are not training from scratch to follow instructions, but rather distilling the general-purpose instruction-following capability from a teacher model that natively possesses instruction-following capability. By freezing the Teacher model during training and employing per-token distillation of its output, the training objective ensures that InstEmb can acquire the Teacher model's instruction following ability.
> > >
> > > We hope these additional explanations and experimental results have fully addressed your concern regarding the model's instruction-following generalization. Thank you again for your insightful questions, which have helped us strengthen our presentation.

---

> > > > ### Comment · Reviewer_xd92 · 2025-11-26
> > > >
> > > > Thank you for your reply! I have adjusted my score based on the new results provided.

---

### Author Response · Authors · 2025-12-01
**summary**

We extend our sincere gratitude to all reviewers and chairs for their invaluable time, insightful feedback, and recognition of our work. Below is a summary of the key revisions made in response to the comments:
# summary of the key revisions
## Additional Experiments
- **Sensitivity Analysis on Look-Ahead Tokens**:

    We added experiments testing different numbers of look-ahead tokens (0, 1, 4, 8), showing performance gains are driven by their presence rather than quantity (added as **new Section 5.2 in revision**).
- **Instruction Robustness Evaluation**:

    We conducted instruction generalization tests following InBedder’s protocol, demonstrating InstEmb’s superior robustness to instruction variations (added in **Appendix A.4 in revision**).
- **Extended Baselines and Benchmarks**:

    We incorporated **new results on benchmark InfoSearch** from Zhou et al. (ICLR 2025) in Table 1. We added a **new baseline PonTE** from Yamada & Zhang (arXiv:2504.16411) in Table 2, and discussed **extra related works** in the Related Work section.

## Conceptual Clarifications
- **Ablation Study on Component Contributions**:

    We clarified the individual effects of look-ahead tokens, representation distillation, and contrastive learning through quantitative comparisons in the existing Table 3.

- **Explicit Definitions of Semantics**:

    We formally defined "primary semantics" (input + instruction) and "complementary semantics" (output) in the** introduction (Lines 43–51)**.

- **Teacher-Student Configuration Justification**:

    We clarified that using an isomorphic teacher rather than a larger one ensures alignment with the student’s own future autoregressive representations, distinguishing it from traditional distillation.

## Presentation Improvements
- **Figure Readability**:

    We enhanced font sizes and clarity in Figures 1 and 3 (formerly Figure 2).
- **Metric Explanations**:

    We added descriptions of nDCG@5 and corrected typographical errors (e.g., "decode" → "decoding", "strongity" → "strength").

We believe above revisions have significantly strengthened the paper’s clarity, rigor, and completeness. Thank you all once again for your constructive guidance and support throughout this process.

# Additional notes
In the end, we provide a brief summary of each reviewer's concern and decision following the previous rebuttal.

- Reviewer xd92: We have mainly addressed reviewer xd92's concerns regarding the number of look-ahead tokens in the paper, the selection of the teacher model, the relative contribution of each loss, the robustness of instruction changes, as well as the lack of benchmarks, baselines, and related work. **Reviewer xd92 ultimately decided to increase the score**.

- Reviewer BmMX: BmMX raised major doubts about the novelty and performance of our paper. We explained the novelty of our paper based on practices in the same field and concurrent work, supplemented the definitions of core concepts, and added explanations regarding the necessity of the joint use of semantics based on the existing Table 5 and the new section 5.2. Finally, we clarified that the work he provided, "Don’t Reinvent the Wheel: Efficient Instruction-Following Text Embedding based on Guided Space Transformation," cannot be directly used as a baseline because its evaluation criteria are inconsistent with previous work (InBedder) and it is not open-sourced. **Reviewer BmMX did not reply to our rebuttal in the end, but we responded to each of his questions one by one**.

- Reviewer GAWy: GAWy's concerns mainly focus on the selection of the teacher model and the presentation of the paper. We have redrawn several images in the paper and corrected the expression errors, and **Reviewer GAWy decided to maintain the original positive score**.

---

### Meta-Review · Area_Chair_ki7S · 2025-12-21

**Summary:**

This submission proposes InstEmb, an instruction-following embedding framework that integrates look-ahead tokens, representation distillation, and contrastive learning to capture both input-side and output-side semantics in a single forward pass. Reviewers generally agree that the approach is technically sound, efficient, and empirically competitive on a range of instruction-aware embedding benchmarks. However, across reviews, concerns consistently center on the incremental nature of the contribution, insufficiently grounded semantic claims, and limited empirical evidence for the central hypothesis that jointly modeling “primary” and “complementary” semantics provides benefits beyond existing output-centric embedding methods.

While the rebuttal adds useful clarifications, additional ablations, and new experiments, it does not fully resolve the core concerns around conceptual novelty and explanatory depth. As a result, despite reasonable empirical performance, the paper falls short of the bar for acceptance in its current form.

**Reviewer Concerns:**

**Reviewer xd92**

Initial concerns focused on: (1) Lack of clear disentanglement between SFT, distillation, and contrastive components; (2) Fixed look-ahead token length; (3) Teacher–student configuration; (4) Ambiguity of treating queries as instructions; (5) Missing relevant baselines and benchmarks

After rebuttal: The authors provided additional ablations, a look-ahead length sensitivity study, instruction robustness experiments, and added several missing benchmarks and discussions. The reviewer explicitly acknowledged these additions and raised their score.

***AC assessment:*** Most technical and experimental concerns were substantively addressed. This reviewer became mildly positive and is not a blocker.


**Reviewer BmMX**

Primary concerns: (1) The work combines well-known techniques, resulting in incremental novelty; (2) Key semantic notions (“primary” vs. “complementary”) were initially vague; (3) The joint-usage hypothesis (that both semantics must be used together) was not convincingly demonstrated; (4) Missing comparison with a highly relevant recent baseline

After rebuttal: (1) Definitions were clarified. (2) The authors reframed novelty around a new optimization objective (“single-pass future semantic preview”), which is coherent but largely conceptual rather than empirical. (3) Evidence for joint usage remains indirect, relying on comparisons to existing pooling strategies rather than a clean factorial ablation. (4) The baseline omission was justified but not fully resolved.

***AC assessment:*** Some concerns were addressed, but the central novelty and hypothesis validation issues remain only partially resolved. The reviewer remains neutral but unconvinced.


**Reviewer GAWy**

Primary concerns: (1) Weak grounding and operationalization of semantic claims; (2) Limited insight into what information is captured by each representation; (3) Presentation issues (figures, metrics explanation); (4) Lack of qualitative or interpretive analysis

After rebuttal: (1) Presentation issues were improved; (2) Semantic distinctions were clarified, but largely by restating alignment objectives. The reviewer explicitly stated they would maintain their original assessment.

***AC assessment:*** Core concerns about explanatory depth and semantic interpretability remain unresolved. This reviewer remains lukewarm and does not see the rebuttal as materially changing the paper’s contribution.

**Reviewer Scores:**

Reviewer xd92:
Likely increased from marginally below the acceptance threshold to marginally above the acceptance threshold based on new experiments.

Reviewer BmMX:
Likely unchanged; remains borderline with persistent novelty concerns.

Reviewer GAWy:
Explicitly unchanged; maintains original assessment.

---

### Decision · Program_Chairs · 2026-01-26

Reject